# Integrated Genomic and Phenotypic Analysis of *Bacillus safensis* LG01 Highlights Its Prospects in Biotechnology and Biocontrol

**DOI:** 10.3390/microorganisms13112605

**Published:** 2025-11-15

**Authors:** Lijuan Yang, Yan Shuai, Jie Ren, Yiqin Yang, Zhou Jiang, Yongjun Lu, Zhenhuang Ge

**Affiliations:** Run Ze Laboratory for Gastrointestinal Microbiome Study, School of Life Sciences, Sun Yat-sen University, Guangzhou 510275, China; yanglj68@mail.sysu.edu.cn (L.Y.); shuaiyan@sysucc.org.cn (Y.S.); renjie20011024@sjtu.edu.cn (J.R.); yangyq78@mail2.sysu.edu.cn (Y.Y.); jiangzh75@mail2.sysu.edu.cn (Z.J.); luyj@mail.sysu.edu.cn (Y.L.)

**Keywords:** *Bacillus safensis*, comparative genomics analysis, biocontrol

## Abstract

*Bacillus safensis* strains have emerged as versatile microbial platforms for bioproduction, combining the benefits of probiotic utility and biocontrol. In this study, we describe the isolation and in-depth characterization of a previously unreported *B. safensis* strain, LG01. The genome of this strain comprises a circular chromosome encoding 13 secondary metabolite biosynthetic gene clusters, 144 carbohydrate-active enzymes, 2 antibiotic resistance loci, and 1 prophage region, indicative of strong antimicrobial and metabolic capacity. Its protein secretion systems support nutrient acquisition, colonization, quorum sensing, and antibiotic synthesis. Our phenotypic assays confirmed the antifungal and antibacterial activity, proteolytic and cellulolytic functions, and robust biofilm formation of the strain. By performing a comparative genomic analysis, we identified 78 strain-specific genes enriched in the bacteriocin immunity and sporulation pathways. Signals of positive selection in the membrane and transcriptional regulator genes further reflect the adaptive evolution underlying the strain’s ecological fitness. Together, these findings advance our understanding of the genomic features of *B. safensis* LG01 and highlight its promise as a candidate for biocontrol and probiotic applications.

## 1. Background

*Bacillus* species are genetically diverse and can thrive in various environments [1]. *Bacillus safensis* is a Gram-positive bacterium first isolated and reported in 2006. The DNA sequence similarity of its *gyrB* gene was only 54–66% and 91.2% when compared to previously known species, leading to its ultimate classification as *B. safensis* [2]. This strain has many desirable properties and wide application prospects; for example, genomic characterization of *B. safensis* RP10, which was isolated from Atacama Desert soil, revealed its adaptations to high heavy metal concentrations, saline conditions, and limited carbon/energy sources. These findings suggest its potential for bioremediation of metal-contaminated environments and enhancing plant stress resilience [3]. These findings suggest its potential for bioremediation of metal-contaminated environments and enhancing plant stress resilience. When sent to the International Space Station, contrary to other strains, *B. safensis* JPL-MERTA-8-2 showed a 60% improvement in growth compared to on Earth [4], reflecting its strong stress resistance. *B. safensis* DVL-43 produces a novel and organic solvent-stable lipase [5], while *B. safensis* CFA-06 can produce catalases and oxidoreductases to degrade aromatic compounds and petroleum aromatic fractions [6], reflecting its strong degradation effect on oil. A mutant *B. safensis* LAU13 isolated with ultraviolet radiation was found to secrete highly active keratinase to degrade chicken feathers without damaging the skin [7]. *B. safensis*, as a rhizobacteria, can reduce abiotic stresses and thus promote plant growth [8], which would be extremely useful tools in sustainable agriculture. An RNase produced by *B. safensis* RB-5 showed antiproliferative activities towards many human transformed cells, and widely retain normal cells [9], surfactin produced by *B. safensis* F4 showed an antitumor activity against T47D breast cancer cells and B16F10 mouse melanoma cells [10], reflecting its potential to be developed into new antitumoral drugs. *B. safensis* S7, and *B. safensis* N4 have a strong metabolic effect on heavy metal ions in soil and thus provided a basic reference for remediation of heavy metal contaminated soil by bacteria. Furthermore, *B. safensis* also possess strong antimicrobial ability. *B. safensis* strains demonstrate significant potential as biocontrol agents and sources of next-generation biopesticides. Strain STJP (NAIMCC-B-02323) exhibits strong antagonism against the phytopathogen *Alternaria alternata* [11], while strain B21 produces the antifungal compounds iturin A2 and iturin A6, which impair hyphal growth in Magnaporthe oryzae by altering membrane permeability [12]. Additionally, strain C3 shows antibacterial activity against *Escherichia coli* and *Pseudomonas syringae* [13], and strain F4 displays broad-spectrum efficacy against several pathogenic strains [10]. *B. safensis* therefore represents a promising biological alternative to synthetic pesticides for sustainable plant disease management.

In this study, we analyzed a novel *B. safensis* strain, LG01, for its potential probiotic and biocontrol properties. Our genomic analysis identified the biosynthetic gene clusters for antimicrobial compounds, with experimental validation confirming both functional traits. Additionally, comparative genomic studies and whole-genome sequencing were conducted to determine which of the four subclusters the orthologous and specific genes of *B. safensis* belonged to. This work sheds light on the evolutionary processes of *B. safensis* isolates and provides a solid foundation for utilizing LG01 as a new biocontrol and probiotic agent, contributing to the current limited literature on this strain.

## 2. Materials and Methods

### 2.1. Bacterial Strain and Cultivation Conditions

Strains were identified through polyphasic characterization, combining cultural, morphological, and molecular analyses. The complete genome sequence has been deposited in GenBank under accession number CP109651.1. For routine cultivation, strains were grown in LB medium at 37 °C for 20 h under microaerophilic conditions. Long-term storage was maintained at −80 °C.

### 2.2. DNA Extraction, Sequencing, and Genome Assembly

Genomic DNA extraction, sequencing and assembly were performed according to established protocols [14]. Genomic DNA was extracted following the manufacturer’s standard protocol for Oxford Nanopore Technologies sequencing platforms (Oxford Nanopore Technologies, Oxford, UK) [15]. DNA quality was assessed using a Qubit 2.0 Fluorometer (Life Technologies, Carlsbad, CA, USA) and NanoDrop system (ThermoFisher Scientific, Wilmington, DE, USA) following manufacturer-recommended protocols. Next, we utilized the standard protocols for the Illumina second-generation high-throughput sequencing platform (Illumina, San Diego, CA, USA) to sequence the total DNA of the samples and used Mecat2 to optimize the original data, obtaining the more accurate third-generation sequencing data. To assemble the 395,699 filtered subsequences with an average length of 2105.4 bp, the SMRT Link v5.1.0 software with default parameters was used for pure third-generation sequencing data, while the Unicycle v0.9.4 software was used for data consisting of data from both second- and third-generation sequencing [16]. Lastly, in order to attain a more accurate genome, we compared the original data and the assembled genome for further correction of errors using Arrow v19.0.0 software [17]. Plasmid sequences were identified through a two-step computational process: unmapped reads were first extracted from whole-genome alignment data using SAMTOOLS v1.19.2, followed by de novo assembly of these reads with SPAdes v4.1.0 software [18].

### 2.3. Functional Annotation

To delineate the functional repertoire of the genome, a systematic functional annotation was conducted [14]. Gene prediction was performed using Glimmer3 v3.02 or Prodigal v2.6.3 for genomes with GC content exceeding 70%, while tRNA identification was carried out with tRNAscan-SE v1.3.1 [19]; and using the rRNAmmer v1.2 was used for identifying the rRNAs [20]. sRNA predictions were first aligned against the Rfam database and then filtered using the cmsearch program. Genomic islands were predicted using IslandPath-DIMOB [21], CRISPR elements with CRISPRdigger [22] and prophages with PHAST [23]. Gene annotation of LG01 was performed through sequence homology analysis using BLAST v2.17.0 and DIAMOND v2.1.9 against multiple databases: NCBI Nr, Swiss-Prot, COG, KEGG, GO, Pfam, CAZy, PHI, VFDB, and CARD. Gene Ontology functional classification was performed using Blast2GO v2.5 [24]. Protein family (Pfam) database annotation based on the Pfam was applicable with a hmmer [25]. In addition, the superior annotation was annotated on the basis of the following databases: Carbohydrate-Active enZYmes Database (CAZyme Database), Pathogen Host Interactions Database (PHI Database), Virulence Factors of Pathogenic Bacteria Database (VFDB Database), Comprehensive Antibiotic Resistance Database (CARD Database). To identify secreted proteins containing signal peptides at the N-terminus rather than transmembrane helices, SignalP, LipoP, TMHMM PSORTb and other software were used [26]. The Circos v0.66 software was used for visual display to obtain the whole genome cycle map [27]. AntiSMASH v5.0.0 was used to predict the gene clusters of secondary metabolites [28]. Phylogenetic analysis was conducted with MEGA X and iTOL using the complete genome sequences of *B. safensis* strains, with all analyses performed under default parameters [29,30].

### 2.4. Antimicrobial Activity and Probiotic Properties

Antibacterial activity was determined using an agar well diffusion method according to established protocols, while probiotic properties were evaluated [31]. A modified agar well diffusion assay was employed to evaluate protease and cellulase production. Proteolysis was assessed on skim milk agar, while cellulose degradation was determined on CMC-Na plates followed by Congo red staining. Biofilm formation was quantified via crystal violet staining. We inoculated 200 µL of an overnight culture into a 96-well plate and removed planktonic cells with PBS washing. The adherent biofilms were then fixed with methanol, stained with crystal violet, and solubilized in glacial acetic acid for absorbance measurement at 595 nm, with a blank medium serving as the negative control. The antibiotic susceptibility of *B. safensis* was assessed against ten different antibiotics using a standard disk diffusion assay. Briefly, bacterial suspensions were spread on antibiotic-free LB agar plates (100 μL per plate). Three antibiotic-impregnated disks and three sterile filter disks (as controls) were evenly spaced on each plate. After 24 h of incubation at 37 °C, the diameters of the inhibition zones were measured. PathogenFinder v1.1 was used to predict the human pathogenicity of *B. safensis* LG01 from its whole-genome sequence.

### 2.5. Comparative Genomic Analysis

Comparative genomics was applied to delineate the pan-genome structure and identify lineage-specific genes [14]. DNA alignments and phylogenetic trees were created using the *gyrB* genes from 13 *B. safensis* strains that underwent complete genome sequencing [32]. The phylogenetic tree was reconstructed using the Maximum Likelihood method, with the strains divided into four classes for comparative analysis. The pan-genome analysis in this study was based on strains selected from the NCBI database. Sequence comparisons were conducted using a 95% megablast identity threshold with standard parameters. For synteny visualization, TBtools v1.09854 was employed with its default settings [33]. Orthologous clusters were delineated across the four strain classes through OrthoVenn2 analysis, applying classification thresholds of 1 × 10^−2^ for E-value and 1.5 for inflation value [34]. All-against-all protein alignments were performed with DIAMOND v0.9.24 using a BLASTX E-value cutoff of 10^−3^ to refine orthologous relationships. Functional characterization of the identified orthologous clusters was completed through Gene Ontology annotation using the AmiGO2 database (http://geneontology.org, accessed on 11 June 2025).

## 3. Result and Discussion

### 3.1. Genome Sequencing and Assembly

*B. safensis* LG01 was sequenced and assembled to obtain integrated genomic information, followed by nanopore sequencing to acquire raw data. Following quality control with Mecat2, 0.83 Gb of high-quality data was retained for assembly, yielding a complete circular chromosome of 3,664,251 bp (Figure 1). The genome has a GC content of 41.73% and encodes 3836 predicted genes, along with 81 tRNAs, 24 rRNAs, 6 sRNAs, one prophage region, and 5 genomic islands (Table 1). No CRISPR arrays or plasmids were detected.

### 3.2. Functional Annotation

Chromosomal COG analysis identified 215 genes of unknown function, while the annotated gene set was dominated by ten functional categories including amino acid transport and metabolism, transcription, and carbohydrate transport and metabolism (Figure 1). To functionally characterize the predicted genes of *B. safensis* LG01, we performed Gene Ontology (GO) and Kyoto Encyclopedia of Genes and Genomes (KEGG) enrichment analyses. The GO analysis revealed significant enrichment in three principal categories: biological processes (e.g., metabolic, cellular, and single-organism processes), cellular components (e.g., membrane and cell parts), and molecular functions (e.g., catalytic activity, binding, and transporter activity) (Figure 2A). The KEGG pathway analysis further indicated strong enrichment in ABC transporters and amino acid biosynthesis, with additional representation in two-component systems and carbon metabolism (Appendix A).

### 3.3. CAZyme Repertoire and Functional Analysis

Carbohydrate-active enzymes (CAZymes) facilitate the synthesis, modification, and degradation of complex carbohydrates [35]. They are categorized into six major classes according to the sequences and structural features of their functional domains. We located a total of 144 genes from the CAZymes family in the LG01 genome sequence. Glycoside hydrolases (30.56%) and carbohydrate esterases (24.31%) were the most dominant, followed by glycosyl transferases (22.22%), carbohydrate-binding modules (18.06%), auxiliary activities (3.47%), and polysaccharide lyases (1.39%) (Figure 2B). CAZymes serve as key genomic markers of an organism’s adaptation to specific ecological niches [36]; of these, glycoside hydrolases catalyze the cleavage or rearrangement of glycosidic bonds in diverse glycoconjugates and polysaccharides, while carbohydrate esterases remove ester-linked decorations from carbohydrates through de-O- or de-N-acylation [37]. By stripping acyl groups from polysaccharides, these esterases not only facilitate subsequent hydrolysis by glycoside hydrolases but also enhance overall biomass saccharification efficiency [38]. The range of CAZymes identified in the present study likely underlies the environmental resilience of the strains examined.

### 3.4. Secondary Metabolism Gene Cluster Analysis and Biocontrol Functions

Studies have shown that *B. safensis* possesses inhibitory capabilities and should be considered as a biocontrol agent for fungi [12,39]; it has also been investigated for its strong inhibitory effect on a variety of bacteria [40]. Recently, *B. safensis* has been shown to harbor a broad spectrum of antimicrobial substance-coding genes in its genomes [41]. In this study, 13 gene clusters associated with the biosynthesis of secondary metabolites, which are significant for the biocontrol capacity against pathogens, were discovered through genome mining of *B. safensis* LG01 (Table 2). Genome annotation revealed that *B. safensis* LG01 encodes diverse biosynthetic pathways for specialized metabolites, including terpenes, β-lactones, RRE-element-containing and linear azol(in)e-containing peptides, sactipeptide, ranthipeptide, and siderophore. In LG01, gene clusters for *Bacillus subtilis* subsp. *subtilis* str. 168 [42], *Bacillus licheniformis* DSM 13 [43], *Bacillus velezensis* FZB42 [44], *Halobacillus halophilus* DSM 2266, *Bacillus pumilus* ATCC 7061, and *Bacillus subtilis* subsp. *spizizenii* ATCC 6633 demonstrated high homology, suggesting the possibility that secondary metabolism gene clusters underwent horizontal transfer between LG01 and these strains [45]. Furthermore, six additional gene clusters encoded enzymes whose functions were not yet known, suggesting that these gene clusters of LG01 are specific to the strain.

*B. safensis* LG01 displayed broad-spectrum antimicrobial activity, inhibiting representative fungal and Gram-positive bacterial pathogens. Notably, it also showed significant inhibition against the Gram-negative pathogen *Legionella pneumophila*, although it was inactive against other Gram-negative strains such as *Escherichia coli* and *Pseudomonas aeruginosa* (Table 3). *L. pneumophila*, when inhaled via contaminated aerosols, evades host immunity by replicating within alveolar macrophages and ultimately causes cell death, a key mechanism in legionellosis pathogenesis [46]. Elucidating the inhibitory mechanism of *B. safensis* LG01 against *L. pneumophila* may offer a novel strategy for combating infections. The selective activity against *L. pneumophila* could be attributed to the action of specific antimicrobial compounds, such as lipopeptides (surfactin), which are known to disrupt membrane integrity. However, the lack of activity against *E. coli* and *P. aeruginosa* may be due to the formidable outer membrane barrier of these bacteria, which acts as a primary defense against macromolecules, coupled with the potential role of efficient efflux pump systems (particularly in *P. aeruginosa*) that can expel toxic compounds. We emphasize that the potent inhibition of *L. pneumophila* a significant human pathogen positions LG01 as a highly promising candidate for the development of narrow-spectrum anti-Legionella agents, thereby potentially mitigating the resistance issues associated with broad-spectrum antibiotic use. Elucidating the inhibitory mechanism of *B. safensis* LG01 against *L. pneumophila* may offer a novel strategy for combating infections.

### 3.5. Antibiotic Resistance

The ubiquity of Bacillus species in the environment means that antibiotic resistance in this genus poses a potential health risk [47]. In bacteria, antibiotic resistance evolves through two principal genetic strategies: target gene mutation and horizontal acquisition of resistance determinants [48]. The intrinsic resistance genes identified in LG01 are chromosomally encoded. The presence of these chromosomally encoded resistance genes likely enhances the strain’s ecological fitness, improving its ability to successfully colonize specific niches where these antibiotics are present.

Horizontal gene transfer, the non-hereditary passage of genetic material between organisms, significantly contributes to bacterial genome plasticity and adaptive evolution [49]. Obtaining foreign genes can effectively alter the genotype of a bacterium, potentially resulting in the development of new traits or even new species [50]. Genes acquired through horizontal gene transfer (HGT) are frequently localized within integrons and mobile genetic elements (MGEs), including transposons, insertion sequences, genomic islands, phages, plasmids, and integrative and conjugative elements (ICEs) [51]. As an important vector for the transfer of resistance genes between bacteria, plasmids play an important role in the global spread of antibiotic resistance [52]. However, there is no plasmid in *B. safensis* LG01 (Figure 1). However, no plasmids were identified in *B. safensis* LG01 (Figure 1); this reduces the risk of using this strain in humans, as these genes were present on genomic DNA rather than plasmid DNA [53].

Previous authors identified the *B. subtilis* subsp. *subtilis* strain RK with a novel class of aminoglycoside transferase (AHP5), which is responsible for resistance against aminoglycoside-based antibiotics [47]. In *Staphylococcus aureus*, MFS transporters actively export structurally different antimicrobial agents from the cells [54]. Our genomic analysis identified two key antibiotic resistance determinants: a major facilitator superfamily (MFS) transporter gene (mdr) that mediates multidrug efflux, and an undecaprenyl pyrophosphate phosphatase gene (baca) involved in bacitracin resistance through cell wall biosynthesis modulation (Appendix A). These genes may allow LG01 to adapt to different survival conditions, hosts, and environments.

Antibiotic susceptibility testing revealed that *B. safensis* was susceptible to nine of the ten antibiotics tested, including penicillin, ampicillin, ceftriaxone, tetracycline, chloramphenicol, gentamicin, erythromycin, ciprofloxacin, and trimethoprim-sulfamethoxazole. No zone of inhibition was observed around the lincomycin (2 μg) disk. Quantitative measurements of the inhibition zones confirmed diameters of 20–40 mm for the nine effective antibiotics (Appendix A). These results demonstrate that the strain exhibits high susceptibility to antibiotics and lacks broad-spectrum resistance, indicating that it can be effectively controlled in practical applications and is unlikely to contribute to the spread of antibiotic resistance.

Analysis of the *B. safensis* LG01 genome using the tool PathogenFinder v1.1 predicted that 33 out of its 3659 coding sequences belong to non-pathogenic families, with no sequences identified as part of pathogenic lineages. These results indicate that *B. safensis* LG01 is non-pathogenic to humans.

### 3.6. Genomic Islands

Genomic islands (GIs) horizontally acquired chromosomal segments typically spanning 10–500 kbp [49], serve as valuable markers for comparative genomics and the identification of strain-specific adaptive genes [1]. Within the genomic architecture of *B. safensis* LG01, we identified five such GIs (Appendix A). GI region 1 comprised 21 coding sequences (CDSs), including an ABC-2 type transport system-associated protein, aminotransferase, mutase, endoribonuclease, permease, and transcriptional regulator. GI region 2 contained 11 CDSs encoding the SMI1/KNR4 family protein, transcriptional regulators, reductase, acetyltransferase, phosphatase, transposase or phage integrase, and phosphotransferase. GI region 3 contained 18 CDSs encoding integrase/recombinase, methyltransferase, flagellar-associated protein, transcriptional pleiotropic repressor, and proteases. GI region 4 contained 27 CDSs, including those encoding synthase, endoribonucleases, oxidoreductases, transcriptional regulators, dehydrogenases, integrase/recombinase, acetyltransferase, DNA helicase, azoreductases, NarL family-associated protein, and glyoxalase/bleomycin resistance protein/dioxygenase. GI region 5 was located in the area from 2,725,696 to 2,738,546 bp, with a length of 12,850 bp; it contained one Alanine tRNA coding gene and 22 CDSs consisting of lipoproteins, transcriptional regulators, amidases, acetyltransferase, and bhlA. BhlA has shown efficacy against multiple drug-resistant bacteria and possesses a broad spectrum of antibacterial activity [55]. GIs confer adaptive advantages to host bacteria through acquisition of novel metabolic pathways, antibiotic resistance determinants, and virulence factors, thereby enhancing fitness and resilience under diverse abiotic stresses [49].

### 3.7. Prophage Elements

Bacteriophages, which multiply passively with their bacterial hosts, are carried by a large number of bacteria and integrate into their genomes as prophages [56]. Lysogeny is ubiquitous and widely distributed throughout the murine gut microbiota [57]. Phages reside in 40–50% of bacterial genomes as prophage elements, which are the primary source of bacterial diversity within species and the main drivers of horizontal gene transfer [58]. Prophage-mediated DNA rearrangement is a common phenomenon in spore-forming bacteria [59], and accurate prophage annotation is therefore essential for complete functional and genomic characterization of bacterial strains [60]. A single prophage region was identified in the LG01 genome (Appendix A), located from 2,723,924 to 2,759,923 bp, with a length of 35,999 bp, and featuring one Alanine tRNA coding gene and 53 CDSs consisting of transcriptional regulators, lipoproteins, integrases, terminases, amidases, phosphatases, bhlA, acetyltransferases, phage portal proteins, putative bacteriophage proteins, transglycosylase, RNA polymerase, and resolvase. These prophage sequences may confer antibiotic resistance, enhance adhesion, and promote environmental adaptation, and they likely play an active role in cell physiology. They also contribute to LG01’s ability to adjust to changes in ecological environments. The functional influence of prophage elements on the diversification of beneficial traits remains uncharacterized.

### 3.8. Host Microbe Secreted Effectors

Surface and secreted proteins are key mediators of host adaptation in bacteria, enabling nutrient acquisition, host cell interactions, and tissue specific colonization [61]. *B. safensis* LG01 encodes an array of surface and secreted proteins that facilitate host bacterium interactions (Appendix A). Key genes implicated in these functions include: *BsLG_04104*, encoding a beta-N-acetylhexosaminidase involved in chitin degradation and glycoconjugate metabolism; *BsLG_01459* [62], encoding a gamma-glutamyltranspeptidase that modifies glutathione derivatives; and *BsLG_00941*, encoding a WxL domain-containing surface protein potentially involved in virulence and host protein interaction [63]. Multiple hydrolytic enzymes including peptidases, subtilisin, pectate lyase, and xylan hydrolase (encoded by genes *BsLG_00300*, *BsLG_02651*, *BsLG_01213*, *BsLG_00711*, *BsLG_01808*, *BsLG_04032*, *BsLG_04364*, *BsLG_04446* and *BsLG_02332*) may support nutrient digestion from complex dietary substrates [64,65]. Furthermore, LG01 harbors global regulators linked to biofilm formation and colonization, such as Spo0A (*BsLG_02744*), wall teichoic acid biosynthesis proteins GtaB (*BsLG_04097*, *BsLG_04094*), and the LuxS/AI-2 quorum-sensing system (*BsLG_03440*) [66,67,68]. Notably, several Rap family phosphorelay-integrating regulators (*BsLG_04580*, *BsLG_02742*, *BsLG_04169*, *BsLG_01426*, *BsLG_02061*, *BsLG_04411*, *BsLG_02106*) were identified, suggesting that in LG01, Rap protein and Phr signaling play prominent roles in modulating quorum sensing and sporulation [69].

### 3.9. Probiotic Properties

The rise of antibiotic-resistant pathogens underscores the need for alternative antimicrobial strategies. *B. safensis* LG01 emerges as a promising probiotic candidate, harboring numerous biosynthetic gene clusters for antimicrobial secondary metabolites (Table 2) and exhibiting potent antibacterial and antifungal activity (Table 3). *B. safensis* VQV8 exhibits potent anti-Vibrio activity and demonstrates safety profiles compatible with probiotic applications [70]. The LG01 strain produces proteases and cellulases which function to degrade dietary proteins and cellulose, consistent with the growth-promoting effects observed in the enzyme-producing strain NPUST1 in tilapia models [71]. LG01 also forms structured biofilms that promote host colonization and ecological persistence (Table 4), while also potentially excluding pathogens as demonstrated in other *B. safensis* strains [72]. Biofilms provide microbes with adaptive advantages by facilitating nutrient acquisition and enhancing stress resilience [13]. In related strains, such biofilms contribute to the exclusion of foodborne pathogens like *Listeria monocytogenes* on the surfaces of produce, though the molecular mechanisms underlying this competitive exclusion require further characterization [73].

Our safety evaluation identified two intrinsic antibiotic resistance genes in LG01 (Appendix A). Consistent with this genus profile [70], the VQV8 strain shows susceptibility to common antibiotics [53]. Collectively, these functional and genomic attributes support the potential of *B. safensis* LG01 as a versatile probiotic.

### 3.10. Comparative Genomic Analysis

#### 3.10.1. Phylogenetic Analysis and Genomic Collinearity

The field of molecular phylogenetics focuses on inferring organismal evolutionary history through comparative analyses of molecular data [74], providing answers to various biological questions [75]. 16S rRNA gene sequencing serves as a powerful diagnostic tool, enabling identification of novel pathogens, uncultured bacteria, phenotypically ambiguous strains, and rarely isolated species [76]. This method is not without its drawbacks, especially when it comes to differentiating closely related species. Molecular phylogenetic studies are increasingly utilizing protein-coding genes such as *gyrA* and *gyrB* as molecular chronometers in neighbor-joining trees, complementing the use of 16S rRNA. While 16S rRNA remains effective for delineating deeper-branching taxonomic relationships [32], *gyrA* and *gyrB* sequences exhibit superior discriminatory power for resolving closely related lineages at intra- and intergeneric levels [14]. Notably, gyrB sequence polymorphisms have proven particularly effective in distinguishing phylogenetic groups within the *B. safensis* complex [77,78].

To evaluate the phylogenetic position of LG01, we reconstructed a *gyrB*-based Maximum Likelihood tree with MEGA X, comparing it against other available *B. safensis* genomes [29]. Thirteen *B. safensis* strains were used in this analysis, and the genome of the LG01 strain was compared to every fully sequenced *B. safensis* strain available in the GenBank database (Figure 3). Additionally, we have provided a comparative table (Appendix A) that systematically outlines the unique genomic features and functional attributes of LG01 compared to other closely related Bacillus strains, including type strains and characterized isolates. Phylogenetically segregated into four distinct classes (I–IV), these strains were subsequently subjected to whole-genome comparative analysis using TBtools, revealing conserved syntenic blocks across the taxonomic divisions (Appendix A).

*B. safensis* Class I comprises five genomically distinct strains AHB11, LG01, IDN1, BRM1, and the psychrotrophic Antarctic isolate U14-5 from Lake Untersee. Our pairwise comparison identified AHB11 as sharing 1687 collinear blocks with BRM1, 1530 with IDN1, 1718 with LG01, and 2019 with U14-5. BRM1 displayed 1373 collinear blocks with IDN1, 1439 with LG01, and 1748 with U14-5. The IDN1-LG01 comparison revealed 1307 collinear blocks, while IDN1 shared 1606 blocks with U14-5. Notably, the LG01-U14-5 comparison showed 1823 collinear blocks, representing one of the highest syntenic relationships within this class. Comparative analysis established U14-5 as the genomic cornerstone of Class I, demonstrating the maximum number of collinear blocks with all other members: 2019 with AHB11, 1748 with BRM1, 1606 with IDN1, and 1823 with LG01. This consistent syntenic predominance suggests that U14-5 may represent an evolutionarily conserved lineage within the *B. safensis* complex (Appendix A) and has the highest syntenic relationship with other strains and was also compared based on Average Nucleotide Identity (ANI) (Appendix A). *B. safensis* ZK-1, PgKB20, KCTC 12796BP, and H31R 08 make up Class II. Whole-genome alignment revealed that H31R 08 shares 2561 collinear blocks with KCTC 12796BP, 2760 with PgKB20, and 2672 with ZK-1. KCTC 12796BP exhibited 2682 collinear blocks with PgKB20 and 2602 with ZK-1. The comparison of PgKB20 and ZK-1 showed 2765 collinear blocks, the highest value observed in Class II. Consistently, PgKB20 demonstrated peak syntenic conservation with all other class members: 2760 blocks with H31R 08, 2682 with KCTC 12796BP, and 2765 with ZK-1, establishing it as the genomic reference for this phylogenetic group. Class III, represented by strains F6 and AHB2, showed substantial genomic conservation, with 2665 collinear blocks between them. Class IV, consisting of U41 and U17-1, exhibited 1520 collinear blocks, indicating more divergent genome architecture compared to other classes. These results demonstrate distinct patterns of syntenic conservation across the phylogenetic spectrum, with PgKB20 serving as the genomic anchor for Class II, while Classes III and IV show characteristic levels of genome structural conservation reflective of their phylogenetic distances.

#### 3.10.2. Comparative Genomic Analysis of Four Sub Clusters

Comparative genomic analysis across the four *B. safensis* phylogenetic classes identified 2975 core orthologous clusters, representing the essential genetic repertoire of this species complex. This conserved genomic foundation exhibits clear phylogenetic stratification, with Class I and II strains showing greater orthology conservation than the more divergent Classes III and IV (Figure 4A). Compared to the other three sub clusters, a total of 3047 genes were identified in Class I, with no specific gene clusters and 16 specific singletons, 11 of which were annotated in the GO terms and five of which were not annotated; in Class II, a total of 3425 genes were identified, with one specific gene cluster (including four genes) and 33 specific singletons, 17 of which were annotated in the GO terms and 17 of which were not annotated; in Class III, a total of 3441 genes were identified, with two specific gene clusters (including eight genes) and 59 specific singletons, 38 of which were annotated in the GO terms and 23 of which were not; in Class IV, a total of 3446 genes were identified, with eight specific gene clusters (including 17 genes) and 295 specific singletons, 92 of which were annotated in the GO terms and 211 of which were not (Figure 4B,C).

Gene Ontology enrichment analysis of subclass-specific gene clusters and singletons revealed distinct functional profiles across the phylogenetic classes. In Class I strains, significant enrichment was observed in two primary GO categories: biological processes and molecular functions, indicating specific evolutionary adaptations in these functional domains (Figure 4D). The enriched biological process terms included iron ion homeostasis (GO:0055072), iron ion transport (GO:0006826), antibiotic biosynthetic process (GO:0017000), and peptidoglycan turnover (GO:0009254). The enriched molecular function terms included hydrolase activity (GO:0016787), DNA binding (GO:0003677), oxidoreductase activity (GO:0050664), methyltransferase activity (GO:0008168), and sugar-phosphatase activity (GO:0050308) (Figure 4E), showing that Class I strains have a better capacity for iron ion homeostasis and transport, which likely contributes to their improved resistance to heavy metal detoxification. The maximum metal tolerance of *B. safensis* ST7 to Fe (III) was 250 mg/L, and it was able to grow in the presence of high concentrations of this element [79]. Therefore, Class I is a potential candidate for restoring soil polluted by multiple heavy metals.

In Class II, GO terms were enriched in the following three categories: biological processes, molecular functions, and cellular components (Figure 4D). Gene Ontology analysis revealed functional specialization in Class II strains, with significant enrichment in antibiotic biosynthesis (GO:0017000) and rhamnose catabolism (GO:0019301) for biological processes. Molecular functions were characterized by metal ion binding (GO:0046872), sequence-specific DNA binding (GO:0043565), and serine-type endopeptidase activity (GO:0004252), while cellular components showed predominant localization to the plasma membrane (GO:0005886) (Figure 4F). The significantly enriched antibiotic biosynthetic process likely indicates the production of metabolites with antimicrobial activity in Class II. Many microorganisms are capable of utilizing rhamnose as a carbon source [80]. Thus, these clusters might enhance the environmental adaptation and food acquisition capabilities of the Class II strains, thereby improving their potential as new biological control agents.

In Class III, significantly enriched biological process GO terms included amino acid transport (GO:0006865), regulation of cell shape (GO:0008360), and response to toxic substances (GO:0009636). The significantly enriched molecular function terms included sequence-specific DNA binding (GO:0043565), oxidoreductase activity (GO:0016705), and DNA binding (GO:0003677). Plasma membranes were the only enriched cellular component term (GO:0005886) (Figure 4G). The Class III strains F6 and AHB2 were isolated from chicken feces and beehives, respectively (Appendix A).

In Class IV, GO terms included viral genome integration into host DNA (GO:0044826), transmembrane transport (GO:0055085), inositol catabolic process (GO:0019310), fatty acid biosynthetic process (GO:0006633), lipid A biosynthetic process (GO:0009245), response to arsenic-containing substance (GO:0046685), and RNA catabolic process (GO:0006401). The significantly enriched molecular function terms included sequence-specific DNA binding (GO:0043565), metal ion binding (GO:0046872), oxidoreductase activity (GO:0016491), potassium-transporting ATPase activity (GO:0008556), ATP binding (GO:0005524), and phosphoprotein phosphatase activity (GO:0004721). The significantly enriched cellular component terms included integral component of membrane (GO:0016021) and plasma membrane (GO:0005886) (Figure 4H). The Class IV strains U41 and U17-1 are psychrotrophic bacteria isolated from Lake Untersee in Antarctica (Appendix A). Thus, these clusters might enhance the adaptability of Class IV strains to cold environments.

These quasi-specific genes represent the fundamental genetic determinants that facilitate their adaptation to distinct lifestyles, minimize intraspecific competition, and promote coexistence and proliferation.

## 4. Conclusions

Our study establishes *B. safensis* LG01 as a genomically versatile and functionally robust strain with dual potential for sustainable agriculture and probiotic development. The expanded repertoire of secondary metabolite clusters, coupled with its experimentally validated antimicrobial and enzymatic activities, underscores LG01’s capacity to thrive in competitive environments while exerting beneficial effects. Evolutionary genomic analyses further reveal adaptive mutations that may underpin its ecological success. These insights not only illuminate the genetic foundations of *B. safensis* as an emerging microbial platform but also position LG01 as a promising candidate for future development into targeted biocontrol agents or next-generation probiotics. The limitation of this study is the laboratory data. Further investigation into its functional efficacy in vivo and under real-world conditions will be essential to translate these genomic advantages into practical applications.

## Figures and Tables

**Figure 1 microorganisms-13-02605-f001:**
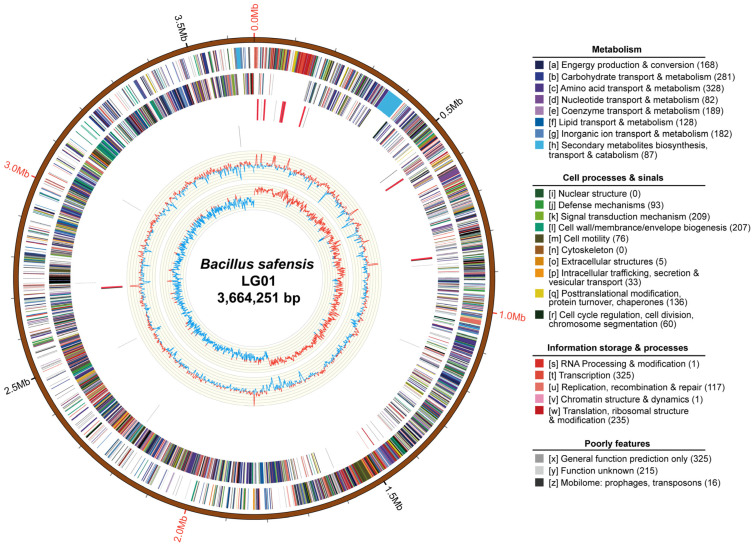
Circular genome map of *Bacillus safensis* LG01. From outer to inner: genomic scale (5 kb intervals); coding sequences (CDSs) on forward (outer ring) and reverse (inner ring) strands, CDSs are colored based on major COG functional categories; tRNA (blue) and rRNA (purple) genes; in the GC content track, light-yellow shading highlights regions where GC content exceeds the genomic average, with peak height reflecting the degree of deviation, blue shading indicates regions below the average; GC skew (dark grey, G > C; red, C > G).

**Figure 2 microorganisms-13-02605-f002:**
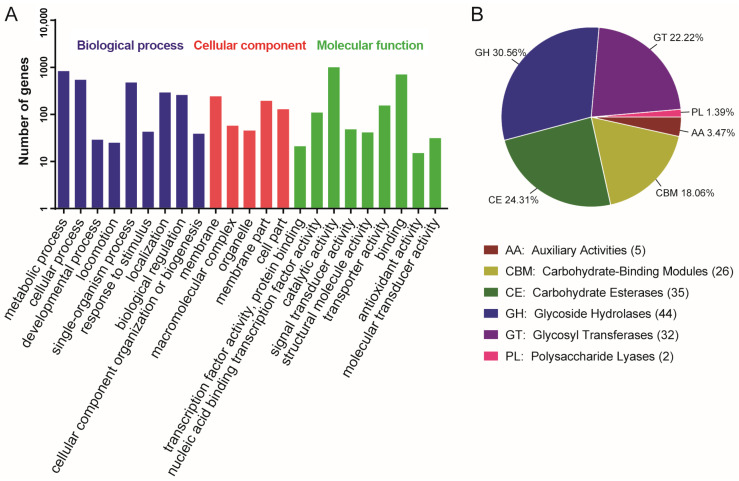
Genome functional annotation of *B. safensis* LG01. (**A**) GO pathway functional Categorization. (**B**) The result of carbohydrate enzyme analysis of *B. safensis* LG01.

**Figure 3 microorganisms-13-02605-f003:**
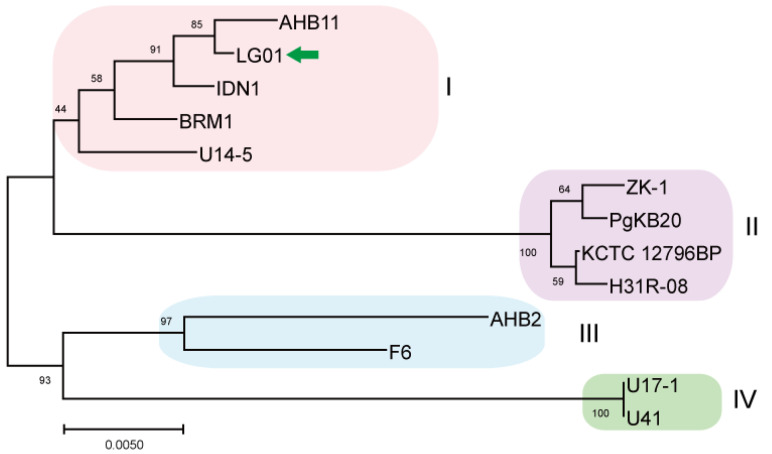
Maximum-Likelihood phylogenetic tree of *B. safensis* LG01 reconstructed from *gyrB* sequences. The dataset for phylogenetic reconstruction comprised 13 nucleotide sequences derived from the complete genomes of available *B. safensis* strains and was conducted using the Maximum Likelihood method. Phylogenetic analysis revealed four subclusters among the 13 strains. The target strain is marked with a green arrow for clarity.

**Figure 4 microorganisms-13-02605-f004:**
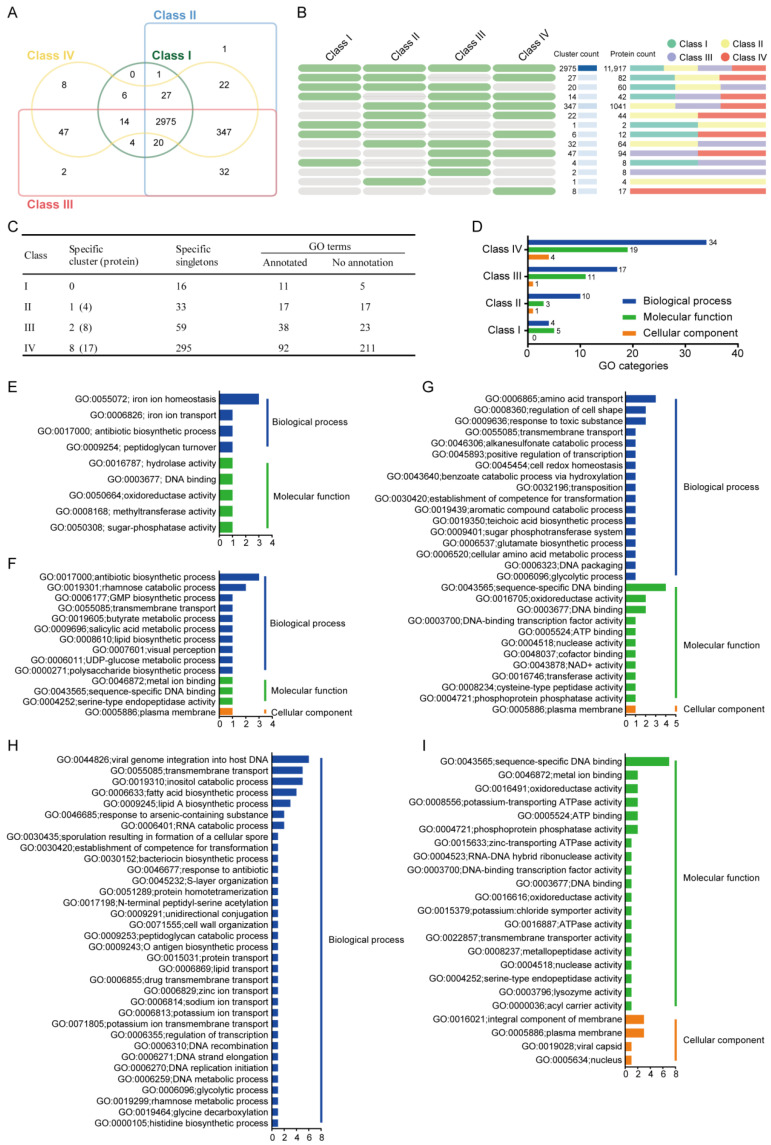
Comparative analysis of orthologous gene clusters across four *B. safensis* subclusters. (**A**,**B**) Venn diagrams illustrate shared and unique orthologous gene clusters. Green cells denote present; gray absent across classes. (**C**,**D**) GO functional annotation of orthologous genes across classes. (**E**–**I**) Class-specific functional profiles for Class I (**E**), Class II (**F**), Class III (**G**), and Class IV (**H**,**I**).

**Table 1 microorganisms-13-02605-t001:** Genome overview of the *B. safensis* LG01 genome.

Features Characteristics	Chromosome
Size (bp)	3,664,251
Number of genes	3836
G + C content (%)	41.73
Genome topology	circular
tRNA	81
rRNA	24
sRNA	6
Prophage	1
Genomic islands	5
CRISPR number	0
plasmid	0
GenBank accession	CP109651.1

**Table 2 microorganisms-13-02605-t002:** Gene clusters involved in synthesis of secondary metabolites in *B. safensis* LG01.

Region	Gene Cluster	Position	Size	Cluster Type ^a^	Metabolite
1	BsLG_00207–BsLG_00237	186,015–208,965	22,951 bp	sactipeptide, ranthipeptide	Sporulation killing factor ^1^
2	BsLG_00397–BsLG_00455	344,928–426,487	81,560 bp	NRPS	Lichenysin ^2^, surfactin ^3^
3	BsLG_01259–BsLG_01291	1,036,827–1,064,980	28,154 bp	terpene, siderophore	Carotenoid ^4^
4	BsLG_01975–BsLG_02001	1,564,132–1,587,295	23,164 bp	RRE-containing, LAP	Plantazolicin ^5^
5	BsLG_02258–BsLG_02289	1,794,694–1,822,344	27,651 bp	betalactone	Fengycin ^3^, mycosubtilin ^6^
6	BsLG_04228–BsLG_04347	3,329,197–3,370,618	41,422 bp	Other	Bacilysin ^3^
7	BsLG_04642–BsLG_04688	3,600,423–3,647,571	47,149 bp	NRPS	Bacillibactin ^1^, paenibactin ^3^
8	BsLG_01028–BsLG_01051	860,675–879,936	19,262 bp	RRE-containing	Unknown
9	BsLG_02111–BsLG_02121	1,682,349–1,691,114	8766 bp	RiPP-like	Unknown
10	BsLG_02362–BsLG_02390	1,889,944–1,911,818	21,875 bp	terpene	Unknown
11	BsLG_02453–BsLG_02502	1,950,022–1,991,122	41,101 bp	T3PKS	Unknown
12	BsLG_02939–BsLG_02956	2,299,648–2,309,974	10,327 bp	RiPP-like	Unknown
13	BsLG_03126–BsLG_03158	2,439,974–2,472,426	32,453 bp	betalactone	Unknown

^a^ NRPS, Non-ribosomal peptide synthetase cluster; RRE-containing, RRE-element containing cluster; LAP, Linear azol(in)e-containing peptides; RiPP-like, Other unspecified ribosomally synthesised and post-translationally modified peptide product (RiPP) cluster; betalactone, beta-lactone containing protease inhibitor; T3PKS, Type III polyketide synthase. ^1^, gene cluster from *Bacillus subtilis* subsp. *subtilis* str. 168; ^2^, *B. licheniformis* DSM 13; ^3^, *Bacillus velezensis* FZB42; ^4^, *Halobacillus halophilus* DSM 2266; ^5^, *Bacillus pumilus* ATCC 7061; ^6^, *Bacillus subtilis* subsp. *spizizenii* ATCC 6633.

**Table 3 microorganisms-13-02605-t003:** Antimicrobial activity of the *B. safensis* LG01.

Categories	Species	Strains	Broth Medium	Antimicrobial Activity ^a^
Fungi				
	*Aspergillus niger*	ATCC 16404	YM	+
	*Candida albicans*	ATCC 10231	YM	+
Gram-negative bacteria				
	*Pseudomonas aeruginosa*	ATCC 27853	LB	−
	*Escherichia coli*	ATCC 25922	LB	−
	*Listeria monocytogenes*	ATCC 19115	BHI	++
	*Ralstonia solanacearum*	GMI1000	CPG	++
	*Legionella pneumophila*	ATCC 33152	AYE	+++
Gram-positive bacteria				
	*Micrococcus luteus*	ATCC 10240	LB	+
	*Staphylococcus aureus*	ATCC 6538	LB	+

^a^ Antimicrobial activity was assessed based on inhibition zone diameters (mm). The size of the zones was categorized as follows: (+) for zones < 10 ± 0.8 mm, (++) for 10–15 ± 1.0 mm, (+++) for >15 ± 0.7 mm (well diameter = 5 mm), and −: No inhibition zone. Data represent mean values from at least three independent experiments.

**Table 4 microorganisms-13-02605-t004:** Probiotic properties of the *B. safensis* LG01.

Biological Process	Activity
**Enzyme function** **^a^**	
Protein degradation (Secreted protease)	+++
Cellulose degradation (Secreted Cellulase)	+++
**Adhesion function** **^b^**	
Formation of biofilm	+++

^a^ Enzyme ability was assessed by measuring the diameter of the degradation zones and scored as follows: (+++) for >15 mm. ^b^ Biofilm formation was evaluated based on ODc (ODc was defined as the mean optical density (OD) of the blank hole plus three standard deviations) and categorized as: strongly adherent (+++) for OD > 4ODc, where ODc represents the mean control value plus three standard deviations. All experiments were performed in at least three biological replicates, and data are presented as mean values.

## Data Availability

The data presented in this study are openly available in [GenBank] at [https://www.ncbi.nlm.nih.gov/], reference number [CP109651.1].

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
