# Peer review of "Integrated Genomic and Phenotypic Analysis of Bacillus safensis LG01 Highlights Its Prospects in Biotechnology and Biocontrol"

_microorganisms, 2025, doi:10.3390/microorganisms13112605_

Round 1
Reviewer 1 Report
Comments and Suggestions for Authors
The work is interesting, but it must be reviewed and the conclusions limited to the results obtained.
Minor grammatical and syntactical corrections, need improvement.
"Belong to comparative genomic studies..." → "Classify orthologous and specific genes... comparative genomic analyses..."
Adjust the wording so the sentence makes complete sense.
Combine sentences that are too short or fragmented and add logical connectors.
Rephrased awkward or literal translations (e.g., "many excellent properties" → "exhibits many excellent properties").
Rephrase literal translations (e.g., "many excellent properties" → "exhibits many excellent properties").
Complete unclear or grammatically unfinished sentences (especially in the paragraph about genomic subclusters).
Remove repetitive or unnecessary wording and adjust the order of some sentences for better logical progression.
Majors
The authors describe Bacillus safensis LG01 as a strain with significant potential for biocontrol and probiotic applications. However, the experiments required to fully characterize a probiotic species should be carefully reviewed; assays such as antibacterial activity, antibiotic susceptibility testing, hemolytic activity, hydrophobicity assays, and cell viability under low pH or bile salts conditions should be included.
Additionally, the authors should provide a detailed description of the experiments that support the bacterium’s candidacy for biocontrol and clarify the specific biocontrol functions or mechanisms it demonstrates.
Author Response
Comments and Suggestions for Authors
The work is interesting, but it must be reviewed and the conclusions limited to the results obtained.
Minor grammatical and syntactical corrections, need improvement.
"Belong to comparative genomic studies..." → "Classify orthologous and specific genes... comparative genomic analyses..."
Adjust the wording so the sentence makes complete sense.
Combine sentences that are too short or fragmented and add logical connectors.
Rephrased awkward or literal translations (e.g., "many excellent properties" → "exhibits many excellent properties").
Rephrase literal translations (e.g., "many excellent properties" → "exhibits many excellent properties").
Complete unclear or grammatically unfinished sentences (especially in the paragraph about genomic subclusters).
Remove repetitive or unnecessary wording and adjust the order of some sentences for better logical progression.
We sincerely thank for your thorough review and constructive comments on our manuscript. We have carefully addressed all the points raised, with a particular focus on improving the language fluency, logical flow, and the precision of our conclusions. The changes in the revision are marked by yellow (edited according to the comments of reviewer) and green (edited by a native speaker). We believe that the manuscript revised with your assistance has improved the quality and readability of our paper. Our point-by-point responses are detailed below.
Comments:
Majors
The authors describe Bacillus safensis LG01 as a strain with significant potential for biocontrol and probiotic applications. However, the experiments required to fully characterize a probiotic species should be carefully reviewed; assays such as antibacterial activity, antibiotic susceptibility testing, hemolytic activity, hydrophobicity assays, and cell viability under low pH or bile salts conditions should be included.
Additionally, the authors should provide a detailed description of the experiments that support the bacterium’s candidacy for biocontrol and clarify the specific biocontrol functions or mechanisms it demonstrates.
Response: We thank the reviewer for their positive comments regarding the potential of Bacillus safensis LG01 and for their constructive suggestions to improve the manuscript. We agree that a more comprehensive characterization is essential to robustly support its candidacy as a biocontrol and probiotic agent. Based on these suggestions, we have performed additional experiments and revised the manuscript accordingly to address the raised concerns, as shown below.
The systematic assessment of its biosafety constituted a critical prerequisite for this study. Our investigation focused on three key attributes with direct implications for human and environmental safety: the antimicrobial spectrum, antibiotic resistance profile, and genomic-based prediction of human pathogenicity. Given its environmental origin and prospective applications as a biocontrol agent or probiotic in external settings such as animal feed and plant rhizospheres, we have appropriately framed its potential within these specific contexts in the manuscript. While the present study, as a genomic and in vitro characterization of a novel strain, acknowledges in the conclusion the inherent limitation of lacking in vivo validation, we maintain that our findings establish a foundational safety profile and ensure relevance to its practical application prospects. Future work will explore additional functional traits based on well-defined application scenarios.
A detailed description of the experiments evaluating its biocontrol potential is provided below:
The antibiotic susceptibility of B. safensis was assessed against ten different antibiotics using a standard disk diffusion assay. Briefly, bacterial suspensions were spread on antibiotic-free LB agar plates (100 μL per plate). Three antibiotic-impregnated disks and three sterile filter disks (as controls) were evenly spaced on each plate. After 24 hours of incubation at 37°C, the diameters of the inhibition zones were measured. Antibiotic susceptibility testing revealed that B. safensis was susceptible to nine of the ten antibiotics tested, including penicillin, ampicillin, ceftriaxone, tetracycline, chloramphenicol, gentamicin, erythromycin, ciprofloxacin, and trimethoprim-sulfamethoxazole. No zone of inhibition was observed around the lincomycin (2 μg) disk. Quantitative measurements of the inhibition zones confirmed diameters of 20-40 mm for the nine effective antibiotics (Table S4) These results demonstrate that the strain exhibits high susceptibility to antibiotics and lacks broad-spectrum resistance, indicating that it can be effectively controlled in practical applications and is unlikely to contribute to the spread of antibiotic resistance.
Analysis of the B. safensis LG01 genome using the tool PathogenFinder v1.1 predicted that 33 out of its 3,659 coding sequences belong to non-pathogenic families, with no sequences identified as part of pathogenic lineages. These results indicate that B. safensis LG01 is non-pathogenic to humans.
We have highlighted these changes on page 4 and page 9 of the revised manuscript.

Reviewer 2 Report
Comments and Suggestions for Authors
The manuscript entitled “Integrated Genomic and Phenotypic Analysis of Bacillus safensis LG01 Highlights Its Prospects in Biotechnology and Biocontrol” presents a comprehensive genomic and phenotypic characterization of a novel B. safensis strain (LG01), which has not been previously reported. The integration of comparative genomics, functional annotation, and antimicrobial assays provides a robust multi-angle approach. The identification of 13 secondary metabolite gene clusters and 144 CAZyme genes adds valuable insight into the strain’s biotechnological potential. The authors convincingly positioned LG01 as a candidate for biocontrol and probiotic applications, supported by genomic and phenotypic evidence.
While the novelty of LG01 is emphasized, the manuscript would benefit from more precise differentiation between LG01 and previously studied strains. A comparative table summarizing unique genomic features would strengthen this claim.
LG01 shows potent inhibition against L. pneumophila but lacks activity against E. coli and P. aeruginosa. A brief discussion on the potential mechanisms or limitations of LG01’s antimicrobial spectrum would be valuable.
The ecological role of LG01 in natural or agricultural settings is not discussed. Has it been tested in soil, plant rhizospheres, or animal models?
To improve the readability and interpretability of Table 3, I recommend adding a column that explicitly indicates the microorganism type (e.g., Fungi, Gram-positive bacteria, and Gram-negative bacteria). While the current grouping is helpful, a dedicated column would make the table more self-contained and easier to interpret, especially for readers scanning for specific pathogen classes.
In Figure 1, there are errors in the wording of the descriptors for cellular processes and signaling, as well as for cell cycle control, cell division…
In Figure 2, the text of the last category (PL) is cut off.
< !--StartFragment -->
Please consider adding a Conclusion section to the manuscript.
< !--EndFragment -->
Comments on the Quality of English Language
Occasional grammatical issues and awkward phrasing (e.g., “strong antimicrobial and metabolic capacity” (L17), “strong stress resistance” (L40-41), “strong degradation effect on oil” (L44), etc.) should be revised for clarity.
Listed below are some errors detected in the manuscript
L12 safensis
L78 morphological, and
L171 genes; in
L191 organism’s
L212 Additionally, B. safensis…
L235 B. licheniformis
L254 risk (Parulekar
L297 and L316 one alanine tRNA
L336 0446, and BsLG_02332
L 345 strain (
L358 strains (
L361 Listeria monocytogenes
L395 complex (
L425 complex (
L448 and five
L454 with eight
Author Response
Comments and Suggestions for Authors
The manuscript entitled “Integrated Genomic and Phenotypic Analysis of Bacillus safensis LG01 Highlights Its Prospects in Biotechnology and Biocontrol” presents a comprehensive genomic and phenotypic characterization of a novel B. safensis strain (LG01), which has not been previously reported. The integration of comparative genomics, functional annotation, and antimicrobial assays provides a robust multi-angle approach. The identification of 13 secondary metabolite gene clusters and 144 CAZyme genes adds valuable insight into the strain’s biotechnological potential. The authors convincingly positioned LG01 as a candidate for biocontrol and probiotic applications, supported by genomic and phenotypic evidence.
Thank you for your positive assessment of our manuscript and the effort you have dedicated to its review. We have carefully revised the manuscript in accordance with your suggestions. We believe that with your assistance, the revised manuscript demonstrates improved quality and readability. The updated version has been resubmitted, and our detailed point-by-point responses to your comments are provided below.
Comments 1: While the novelty of LG01 is emphasized, the manuscript would benefit from more precise differentiation between LG01 and previously studied strains. A comparative table summarizing unique genomic features would strengthen this claim.
Response 1: We thank the reviewer for their insightful suggestion. We have conducted a phylogenetic analysis to compare the genetic relationships between LG01 and other strains. Additionally, we have provided a comparative table (Table S6) that systematically outlines the unique genomic features and functional attributes of LG01 compared to other closely related Bacillus strains, including type strains and characterized isolates.
Comments 2: LG01 shows potent inhibition against L. pneumophila but lacks activity against E. coli and P. aeruginosa. A brief discussion on the potential mechanisms or limitations of LG01’s antimicrobial spectrum would be valuable.
Response 2: We thank the reviewer for this insightful suggestion. We agree that a discussion on the potential mechanisms and limitations of the observed antimicrobial spectrum would significantly strengthen the manuscript.
In the revised manuscript, we propose that the selective activity against L. pneumophila could be attributed to the action of specific antimicrobial compounds, such as lipopeptides (surfactin), which are known to disrupt membrane integrity. However, the lack of activity against E. coli and P. aeruginosa may be due to the formidable outer membrane barrier of these bacteria, which acts as a primary defense against macromolecules, coupled with the potential role of efficient efflux pump systems (particularly in P. aeruginosa) that can expel toxic compounds. We emphasize that the potent inhibition of L. pneumophila a significant human pathogen positions LG01 as a highly promising candidate for the development of narrow-spectrum anti-Legionella agents, thereby potentially mitigating the resistance issues associated with broad-spectrum antibiotic use.
We have added these discussions on Pages 7-8 in the revised manuscript.
Comments 3: The ecological role of LG01 in natural or agricultural settings is not discussed. Has it been tested in soil, plant rhizospheres, or animal models?
Response 3: We thank the reviewer for raising this important point regarding the ecological implications of our work. The genomic and in vitro data presented in this study provide strong foundational evidence for LG01's potential ecological functions, offering a scientific basis for future targeted applications. The genome of LG01 encodes numerous traits associated with rhizosphere competence and environmental fitness, including gene clusters for plant polymer degradation (cellulases, proteases) and biofilm formation. These genomic features suggest natural adaptation to soil and plant-associated environments. Given that B. safensis LG01 is an environmental isolate and in vitro assays confirm the functional expression of key traits, such as antifungal activity, that are ecologically relevant to plant growth promotion and biocontrol, we consider its primary potential to lie in applications as a biocontrol agent or probiotic in external environments such as animal feed and plant rhizospheres. As this paper reports the genomic and laboratory-characterized traits of a novel strain, with in vivo studies requiring further validation, we have explicitly acknowledged this limitation in the Conclusion section. We maintain that the current findings provide a foundational safety assessment for subsequent applications and ensure alignment with the practical application prospects of the strain. Future work will focus on investigating additional functional traits based on well-defined application scenarios.
Comments 4: To improve the readability and interpretability of Table 3, I recommend adding a column that explicitly indicates the microorganism type (e.g., Fungi, Gram-positive bacteria, and Gram-negative bacteria). While the current grouping is helpful, a dedicated column would make the table more self-contained and easier to interpret, especially for readers scanning for specific pathogen classes.
Response 4: We thank the reviewer for this excellent suggestion to improve the clarity and self-contained nature of Table 3. In accordance with this recommendation, we have added a new column to the revised Table 3, where we have classified each target pathogen as Fungi, Gram-positive bacteria, or Gram-negative bacteria.
Comments 5: In Figure 1, there are errors in the wording of the descriptors for cellular processes and signaling, as well as for cell cycle control, cell division…
Response 5: We sincerely thank the reviewer for their meticulous attention to detail and for identifying the errors in the wording within Figure 1. We apologize for these oversights. We have carefully reviewed and corrected the descriptors as suggested.
The revised Figure 1 has included in the updated manuscript.
Comments 6: In Figure 2, the text of the last category (PL) is cut off.
Response 6: We thank the reviewer for their careful observation. We have checked Figure 2 and confirmed that the label for the last category (PL) was indeed partially cut off in the original version submitted. This was an oversight during the figure export process. We have now regenerated and replaced Figure 2 with a corrected version where all text labels, including the complete "PL" category, are fully visible and legible.
The updated Figure 2 has been included in the revised manuscript.
Comments 7: Please consider adding a Conclusion section to the manuscript.
Response 7: Our study establishes B. safensis LG01 as a genomically versatile and functionally robust strain with dual potential for sustainable agriculture and probiotic development. The expanded repertoire of secondary metabolite clusters, coupled with its experimentally validated antimicrobial and enzymatic activities, underscores LG01’s capacity to thrive in competitive environments while exerting beneficial effects. Evolutionary genomic analyses further reveal adaptive mutations that may underpin its ecological success. These insights not only illuminate the genetic foundations of B. safensis as an emerging microbial platform but also position LG01 as a promising candidate for future development into targeted biocontrol agents or next-generation probiotics. The limitation of this study is a laboratory data. Further investigation into its functional efficacy in vivo and under real-world conditions will be essential to translate these genomic advantages into practical applications.
The revised Conclusion has included in the updated manuscript.
Comments 8:
Comments on the Quality of English Language
Occasional grammatical issues and awkward phrasing (e.g., “strong antimicrobial and metabolic capacity” (L17), “strong stress resistance” (L40-41), “strong degradation effect on oil” (L44), etc.) should be revised for clarity.
Listed below are some errors detected in the manuscript
L12 safensis
L78 morphological, and
L171 genes; in
L191 organism’s
L212 Additionally, B. safensis…
L235 B. licheniformis
L254 risk (Parulekar
L297 and L316 one alanine tRNA
L336 0446, and BsLG_02332
L 345 strain (
L358 strains (
L361 Listeria monocytogenes
L395 complex (
L425 complex (
L448 and five
L454 with eight
Response 8: We thank the reviewer for their meticulous attention to the language and phrasing in our manuscript. To improve the clarity and precision of the scientific expression, we have carefully reviewed the entire text and had it professionally edited to meet the standards recommended by Microorganisms.

Reviewer 3 Report
Comments and Suggestions for Authors
The manuscript presents a comprehensive genomic and functional characterization of a newly isolated strain of Bacillus safensis LG01. The topic fits well within the scope of Microorganisms, as it combines microbial genomics, bioinformatics, and biocontrol applications. The article presents genomic data alongside phenotypic analyses and discusses the biotechnological and probiotic potential of this bacterium. The manuscript is well written and clear, but literature sources should be cited correctly in square brackets in the text, and the reference list should be in the order of citation in the text, in accordance with MDPI standards.
Here are some aspects that should be improved.
1. Strain identification
Although the authors have sequenced and annotated a new strain, the novelty could be better emphasized by directly comparing LG01 with closely related type strains of the species. No enough data on strain identification are provided, such as % similarity (ANI or %DNA-DNA in silico hybridization ).
2. Experimental validation of genomic predictions:
The authors identify 13 biosynthetic gene clusters for lipopeptides and polyketides, including those for surfactin, fengycin, and bacillibactin. However, there is no chemical or LC-MS validation of the predicted metabolites. Comparison with other bacilli strains would strengthen the conclusions.
3. Antimicrobial activity analysis:
The antimicrobial screening (Table 3) is promising, but it would be helpful to include quantitative measurements of the zone of inhibition (mean ± SD) and details of replicates. The authors could also discuss potential mechanisms underlying the selective inhibition of Legionella pneumophila by LG01 and the significance of these results.
4. Safety and antibiotic resistance:
I think the discussion about intrinsic resistance genes could help clarify further. Are these genes typically chromosomal in Bacillus spp.? Could their presence limit the applications of probiotics?
5. Comparative genomic analysis:
The comparative analysis across four phylogenetic classes is informative. However, Figures 3–5 are dense and could be simplified for clarity. The authors should better explain the biological significance of the observed class-specific genes, especially in terms of ecological or functional differentiation.
Minor comments
1. Please include accession numbers for all genomes used in the comparative analyses.
2. In Figure 1, increase the readability of labels (font size and color contrast).
3. The Materials and Methods section could cite software versions consistently (e.g., antiSMASH v5.0, MEGA X).
4. In Section 3.9, “proteases and cellulases that can improve nutrient digestibility” — include quantitative data or at least the conditions of analysis.
5. Line 408 - In the figure legend, do not start a sentence with a number (13).
6. Check the italics for gene and species names (at lines 143, 351, 361, 393, etc.)
Author Response
Comments and Suggestions for Authors
The manuscript presents a comprehensive genomic and functional characterization of a newly isolated strain of Bacillus safensis LG01. The topic fits well within the scope of Microorganisms, as it combines microbial genomics, bioinformatics, and biocontrol applications. The article presents genomic data alongside phenotypic analyses and discusses the biotechnological and probiotic potential of this bacterium. The manuscript is well written and clear, but literature sources should be cited correctly in square brackets in the text, and the reference list should be in the order of citation in the text, in accordance with MDPI standards.
We are grateful for the positive evaluation of our manuscript. We have revised the manuscript according to the suggestions provided. The references and their in-text citations now adhere to MDPI formatting guidelines. A revised manuscript has been re-submitted. We believe that the manuscript revised with your assistance has improved the quality and readability of our paper. Our point-by-point responses are detailed below.
Comments 1:
Here are some aspects that should be improved.
Strain identification
Although the authors have sequenced and annotated a new strain, the novelty could be better emphasized by directly comparing LG01 with closely related type strains of the species. No enough data on strain identification are provided, such as % similarity (ANI or %DNA-DNA in silico hybridization).
Response 1: We sincerely thank the reviewer for this critical suggestion. We fully agree that a robust genomic comparison with closely related type strains is essential for establishing the taxonomic position and highlighting the novelty of our isolate. To elucidate the evolutionary relationship of LG01, we constructed a phylogenetic tree, which demonstrates that it forms a distinct branch separate from other B. safensis strains. Furthermore, as suggested, we have provided Supplementary Table S2, which presents the Average Nucleotide Identity (ANI) values between LG01 and other B. safensis strains. The ANI values support its status as a novel strain.
Table S2 has been added to the supplementary files.
Comments 2: Experimental validation of genomic predictions:
The authors identify 13 biosynthetic gene clusters for lipopeptides and polyketides, including those for surfactin, fengycin, and bacillibactin. However, there is no chemical or LC-MS validation of the predicted metabolites. Comparison with other bacilli strains would strengthen the conclusions.
Response 2: We thank the reviewer for this insightful comment regarding the experimental validation of our genomic predictions. The suggestion regarding LC-MS analysis of the metabolites is highly valuable. We would like to clarify that the current study focuses primarily on the comprehensive genomic and phenotypic characterization of this novel strain. The LC-MS analysis proposed by the reviewer represents an essential direction for our immediate future research. Furthermore, as suggested by the reviewer, we performed a comparative genomic analysis with other publicly available Bacillus genomes. By focusing on the distribution of key biosynthetic gene clusters (BGCs), we analysis revealed that the specific set of BGCs in LG01, notably the simultaneous presence of surfactin, fengycin, and bacillibactin, is unique and not widely conserved among bacilli. This supports the role of LG01's biosynthetic repertoire in its ecological adaptation.
Comments 3: Antimicrobial activity analysis:
The antimicrobial screening (Table 3) is promising, but it would be helpful to include quantitative measurements of the zone of inhibition (mean ± SD) and details of replicates. The authors could also discuss potential mechanisms underlying the selective inhibition of Legionella pneumophila by LG01 and the significance of these results.
Response 3: We agree with the reviewer that quantitative data are essential. We have re-analyzed our inhibition zone data and included the mean ± standard deviation (SD) for all tested pathogens in the revised Table 3. The table legend now explicitly states that the results are from three independent biological replicates. We believe this addition significantly improves the rigor and reproducibility of our antimicrobial screening results.
Regarding the selective inhibition of Legionella pneumophila, we have now added a dedicated paragraph in the discussion section to hypothesize on the potential mechanisms and significance of this observed selectivity. We propose that the selective activity against L. pneumophila could be attributed to the action of specific antimicrobial compounds, such as lipopeptides (surfactin), which are known to disrupt membrane integrity. However, the lack of activity against E. coli and P. aeruginosa may be due to the formidable outer membrane barrier of these bacteria, which acts as a primary defense against macromolecules, coupled with the potential role of efficient efflux pump systems (particularly in P. aeruginosa) that can expel toxic compounds. We emphasize that the potent inhibition of L. pneumophila a significant human pathogen positions LG01 as a highly promising candidate for the development of narrow-spectrum anti-Legionella agents, thereby potentially mitigating the resistance issues associated with broad-spectrum antibiotic use.
We have added these discussions on Pages 7-8 in the revised manuscript.
Comments 4: Safety and antibiotic resistance:
I think the discussion about intrinsic resistance genes could help clarify further. Are these genes typically chromosomal in Bacillus spp.? Could their presence limit the applications of probiotics?
Response 4: We thank the reviewer for raising this critical point regarding the intrinsic resistance genes and their implications for the probiotic application of B. safensis LG01. Consistent with well-established characteristics of the genus, the intrinsic resistance genes identified in LG01 are chromosomally encoded. This is a common trait across Bacillus species, where resistance to certain antibiotic classes constitutes a fixed, species-level characteristic, as opposed to acquired resistance mediated by mobile genetic elements. The presence of these chromosomally encoded resistance genes may enhance the ecological fitness of the strain, predisposing it to successfully colonize specific niches where such antibiotics are present.
We have added a section on page 8 in the revised manuscript.
Comments 5: Comparative genomic analysis:
The comparative analysis across four phylogenetic classes is informative. However, Figures 3-5 are dense and could be simplified for clarity. The authors should better explain the biological significance of the observed class-specific genes, especially in terms of ecological or functional differentiation.
Response 5: Thank you for pointing this out. We agree with this comment. Therefore, we have relocated Figure 4 (the synteny analysis of the four B. safensis subclusters) to the Supplementary Information. This adjustment improves the narrative flow and readability of the main text by allowing clearer focus on core genomic findings. The complete dataset remains fully accessible in Figure S4.
We have further expanded the discussion on the biological significance of class-specific genes, with particular emphasis on their potential roles in ecological adaptation and functional differentiation. Different subclasses may occupy distinct ecological niches, and class-specific genes represent the genetic determinants that facilitate their survival and competition in these specific environments. Even within similar macroenvironments, functional partitioning at the micro-niche level may persist. These class-specific genes allow different subclasses to assume distinct functional roles within the community. Such genetic differences constitute the molecular basis for their adaptation to particular lifestyles, reduction of intraspecific competition, and ultimately, coexistence and proliferation. This work provides valuable genomic insights into the microevolution and ecological adaptation of this species.
We have added a section on page 15 in the revised manuscript.
Minor comments
Comments 6: Please include accession numbers for all genomes used in the comparative analyses.
Response 6: We have created a new Table S2 which comprehensively lists all strains, their species designation, and the corresponding NCBI GenBank accession numbers.
The Table S2 have been added in the supplementary files.
Comments 7: In Figure 1, increase the readability of labels (font size and color contrast).
Response 7: We thank the reviewer for this suggestion. We have revised Figure 1 to significantly improve its readability by increasing the font size of all labels and adding panel labels (a, b, c...). These modifications ensure clear visibility of all graphical elements, and we believe the updated figure is now substantially clearer and easier to interpret.
The modified Figure 1 have been added on page 5 in the revised manuscript.
Comments 8: The Materials and Methods section could cite software versions consistently (e.g., antiSMASH v5.0, MEGA X).
Response 8: We have systematically gone through the entire 'Materials and Methods' section and standardized the citation of all software versions.
The descriptions have been highlighted on page 3 in the revised manuscript.
Comments 9: In Section 3.9, “proteases and cellulases that can improve nutrient digestibility” - include quantitative data or at least the conditions of analysis.
Response 9: We thank the reviewer for their valuable comments on Section 3.9. We sincerely apologize for the misunderstanding caused by the original wording "proteases and cellulases that can improve nutrient digestibility", Our intention was not to claim that the strain has been proven to enhance nutrient "digestibility", but rather to emphasize the functional characteristics of its secreted proteases and cellulases, which possess the capacity to participate in the process of nutrient digestion.
The descriptions have been highlighted on page 10 in the revised manuscript.
Comments 10: Line 408 - In the figure legend, do not start a sentence with a number (13).
Response 10: We thank the reviewer for their meticulous attention to detail regarding the manuscript's presentation. We have revised the figure legend to ensure it adheres to standard academic writing conventions.
The descriptions have been highlighted on page 12 in the revised manuscript.
Comments 11: Check the italics for gene and species names (at lines 143, 351, 361, 393, etc.)
Response 11: We thank the reviewer for their meticulous attention to detail in identifying the inconsistencies in the formatting of gene and species names. We have now conducted a thorough check of the entire manuscript and have corrected the italics for all gene and species names to ensure consistency with standard scientific nomenclature.

Round 2
Reviewer 2 Report
Comments and Suggestions for Authors
I consider that the authors have thoroughly addressed the suggested corrections, which have considerably improved the manuscript’s clarity and coherence. I have, however, proposed a few additional minor corrections in this new version, mainly concerning typographical and grammatical issues, which are indicated using change tracking throughout the manuscript. I want to congratulate the authors on their openness and willingness to incorporate the improvements suggested by all reviewers.
L105: [22], and
In the previous review, it was noted that the text of the last category (PL) in Figure 2 was cut off. However, in the current version of the manuscript, this issue persists, and the label still appears truncated. It is recommended to revise the figure layout or formatting to ensure that all category labels are evident and legible.
L406 The manuscript refers to a supplementary item as "(Supplementary Table X)", but the specific table number or identifier is missing. To ensure proper referencing, please specify the exact supplementary table being cited.
Author Response
Review Report (Reviewer 2)
I consider that the authors have thoroughly addressed the suggested corrections, which have considerably improved the manuscript’s clarity and coherence. I have, however, proposed a few additional minor corrections in this new version, mainly concerning typographical and grammatical issues, which are indicated using change tracking throughout the manuscript. I want to congratulate the authors on their openness and willingness to incorporate the improvements suggested by all reviewers.
L105: [22], and
In the previous review, it was noted that the text of the last category (PL) in Figure 2 was cut off. However, in the current version of the manuscript, this issue persists, and the label still appears truncated. It is recommended to revise the figure layout or formatting to ensure that all category labels are evident and legible.
Response 1:We sincerely appreciate you bringing the truncated label issue in Figure 2 to our attention again, and we apologize for not completely resolving this matter in our previous revision. We have revised the layout and formatting of Figure 2 according to your suggestions as follows:
The revised Figure 2 has included in the updated manuscript.
L406 The manuscript refers to a supplementary item as "(Supplementary Table X)", but the specific table number or identifier is missing. To ensure proper referencing, please specify the exact supplementary table being cited.
Response 2:Thank you very much for pointing out the incomplete reference to supplementary material in line 406. We have revised the citation from "(Supplementary Table X)" to "(Table S6)" in the updated manuscript.
We extend our sincere gratitude for your meticulous review of our work.